# Isolation of diborenes and their 90°-twisted diradical congeners

Julian Böhnke[1,2], Theresa Dellermann[1,2], Mehmet Ali Celik[1,2,3], Ivo Krummenacher[1,2], Rian D. Dewhurst [1,2], Serhiy Demeshko[4], William C. Ewing[1], Kai Hammond[1,2], Merlin Heß[1,2], Eckhard Bill[5], Eileen Welz [3], Merle I.S. Röhr [3], Roland Mitrić[3], Bernd Engels[3], Franc Meyer [4] & Holger Braunschweig [1,2]

Molecules containing multiple bonds between atoms—most often in the form of olefins—are ubiquitous in nature, commerce, and science, and as such have a huge impact on everyday life. Given their prominence, over the last few decades, frequent attempts have been made to perturb the structure and reactivity of multiply-bound species through bending and twisting. However, only modest success has been achieved in the quest to completely twist double bonds in order to homolytically cleave the associated π bond. Here, we present the isolation of double-bond-containing species based on boron, as well as their fully twisted diradical congeners, by the incorporation of attached groups with different electronic properties. The compounds comprise a structurally authenticated set of diamagnetic multiply-bound and diradical singly-bound congeners of the same class of compound.

[1] Institute for Inorganic Chemistry, Julius-Maximilians-Universität Würzburg, Am Hubland, 97074 Würzburg, Germany. [2] Institute for Sustainable Chemistry & Catalysis with Boron, Julius-Maximilians-Universität Würzburg, Am Hubland, 97074 Würzburg, Germany. [3] Institute for Physical and Theoretical Chemistry, Julius-Maximilians-Universität Würzburg, Am Hubland, 97074 Würzburg, Germany. [4] Institut für Anorganische Chemie, Universität Göttingen, Tammannstraße 4, 37077 Göttingen, Germany. [5] Max-Planck-Institut für Chemische Energiekonversion, Stiftstraße 34–36, 45470 Mülheim an der Ruhr, Germany. Correspondence and requests for materials should be addressed to H.B. (email: h.braunschweig@uni-wuerzburg.de)

Olefins are ubiquitous in biological systems, crude oil, medicine, and industry, and have been detected in interstellar media, comets, and carbonaceous meteorites[1–3]. Having one degree of electronic unsaturation more than relatively inert alkanes, olefins are the simplest and most earth-abundant hydrocarbons that show reactivity under ambient conditions, leading to their widespread industrial use as well as their implication in questions related to the origins of life[4]. Olefins are molecules that contain a double bond between two carbon atoms consisting of one bond of σ symmetry and one of π symmetry (Fig. 1a, left). This π bond is the root of the countless synthetic uses of olefins, as it contains a concentration of electron density that can interact with a wide range of reagents.

The π orbital of olefins requires the parallel alignment of two carbon $p$ orbitals, and thus also the coplanarity of the two carbon atoms, as depicted in Fig. 1a, left. Since at least the 1960s, a major goal of organic chemistry has been to push the limits of the olefin model through the construction of strained, bent, and twisted alkenes[5,6]. Twisting a (symmetrical) olefin by 90° would completely break its π bond, presumably leading to a triplet diradical (Fig. 1a, right) and making it susceptible to otherwise unfavored cycloaddition reactions[7,8] and further decomposition pathways[9]. This destabilizing response of alkenes to twisting is also the basis of the widely applicable Bredt's rule[10–12], which states that double bonds at the bridgehead positions of small bicyclic systems are disfavored (and reactive) due to their necessarily twisted geometry.

It is thus unsurprising that efforts to isolate "twisted olefin" compounds have resulted in only partial success: isolation of diamagnetic, incompletely twisted structures (up to 66° in the structurally authenticated compound perchloro-9,9′-bifluorenylidene)[13–18]. Beyond carbon–carbon double bonds, sterically hindered heavier olefin analogs such as disilenes (of the form $R_2Si = SiR_2$) also withstand significant twisting of the double bond (up to 55°)[19–23]. In a key 2015 study, Sekiguchi and coworkers reported that one of these disilenes ($Si_2(SiMetBu_2)_4$) was able to thermally populate a triplet state by twisting of the (already highly twisted) Si=Si bond above 77 °C. This diradical compound was characterized by electron paramagnetic resonance (EPR) spectroscopy and calculated to have an average SiSiSiSi dihedral angle of 75.2°; however, its structure could not be authenticated experimentally[22]. This work remains the forefront of chemists' efforts to isolate the diradical products of twisting element–element double bonds.

In this work, we present the isolation of planar, doubly-base-stabilized diborenes (Fig. 1b, left)[24–27], as well as their 90°-twisted diradical congeners (Fig. 1b, right), through the use of different Lewis basic units. The former contain conventional B=B double bonds, while the latter show no B–B multiple bonding and host unpaired electrons in each of the two independent, delocalized π systems. The isolation of the stable ground-state diradical species is facilitated by the outstanding ability of cyclic (alkyl)(amino) carbene (CAAC)[28,29] units to stabilize[30] adjacent radicals[31–34]. This work marks, to the best of our knowledge, the first isolation and structural confirmation of the diradical products of complete homolysis and 90° twisting of double bonds.

## Results

**Synthesis of diborenes and their diradical congeners.** Simple room-temperature (RT) addition of either equal amounts or slight excesses of dibutyldisulfide, diphenyldisulfide, or diphenyldiselenide to solutions of the triple-bond-containing diboryne species $B_2(IDip)_2$ (IDip = 1,3-bis(2,6-diisopropylphenyl)imidazol-2-ylidene) led to new $^{11}B$ NMR signals to higher field of that of the triply-bound precursor ($\delta_B$ 39)[24], signifying the formation of

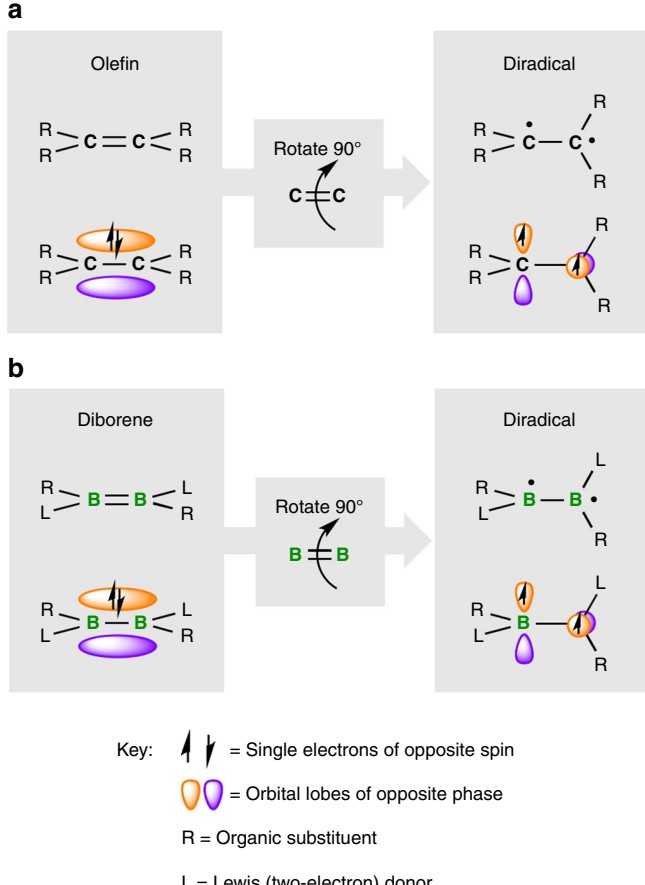

**Fig. 1** Twisting double bonds. Orbital changes accompanying the conceptual twisting of element–element double bonds in alkenes (**a**) and diborenes (**b**) to form diradical species

three new products, **1a–c** (Fig. 2a, top). After short purification steps, **1a–c** were obtained as pure green (**1a**) or purple (**1b,c**) solids with $^{11}B$ NMR signals (**1a**: $\delta_B$ 30; **1b**: $\delta_B$ 25; **1c**: $\delta_B$ 22) in a similar range to those of known doubly NHC-stabilized diborenes ($\delta_B$ 18–28)[24–27], but significantly more downfield than those of two previously observed diboratellurirenium cations (*cyclo*-[Te (R)B(NHC)B(NHC)]⁺) prepared by the analogous reactions of $B_2(IDip)_2$ with diphenylditelluride ($\delta_B$ 0)[35]. Single-crystal X-ray diffraction confirmed the *trans*-diborene nature of **1a–c**, all three molecules displaying effectively planar LRB=BRL double bond units (Fig. 2b, top), and short B=B double bonds (**1a**: 1.585(4) Å; **1b**: avg. 1.568(6) Å; **1c**: avg. 1.565(5) Å).

We also attempted the analogous addition of dibutyldisulfide, diphenyldisulfide, or diphenyldiselenide to solutions of a more cumulenic CAAC analog of $B_2(IDip)_2$ ($B_2(CAAC)_2$; CAAC = 1-(2,6-diisopropylphenyl)-3,3,5,5-tetramethylpyrrolidin-2-ylidene)[36]. However, to our surprise, all three reaction mixtures became very dark, and all attempts to obtain NMR spectra from the mixtures gave either silent spectra or exceptionally broad signals well outside the normal diamagnetic chemical shift ranges. Solvent extraction and recrystallization provided the black solids **2a–c** (Fig. 2a, bottom), with elemental analyses corresponding to 1:1 combination of the reagents, in moderate yields. Single-crystal X-ray diffraction allowed us to elucidate the structures of **2a–c** (Fig. 2b, bottom). In marked contrast to the structures of diborenes **1a–c**, the structures of **2a–c** showed orthogonal C-B-S/Se planes, with S/Se-B-B-S/Se torsion angles near 90° (**2a**: 85.5 (1)°; **2b**: 100.9(1)°; **2c**: 99.7(2)°), and long, single B–B bonds (**2a**: 1.728(2) Å; **2b**: 1.713(2) Å; **2c**: 1.700(4) Å). On either side of

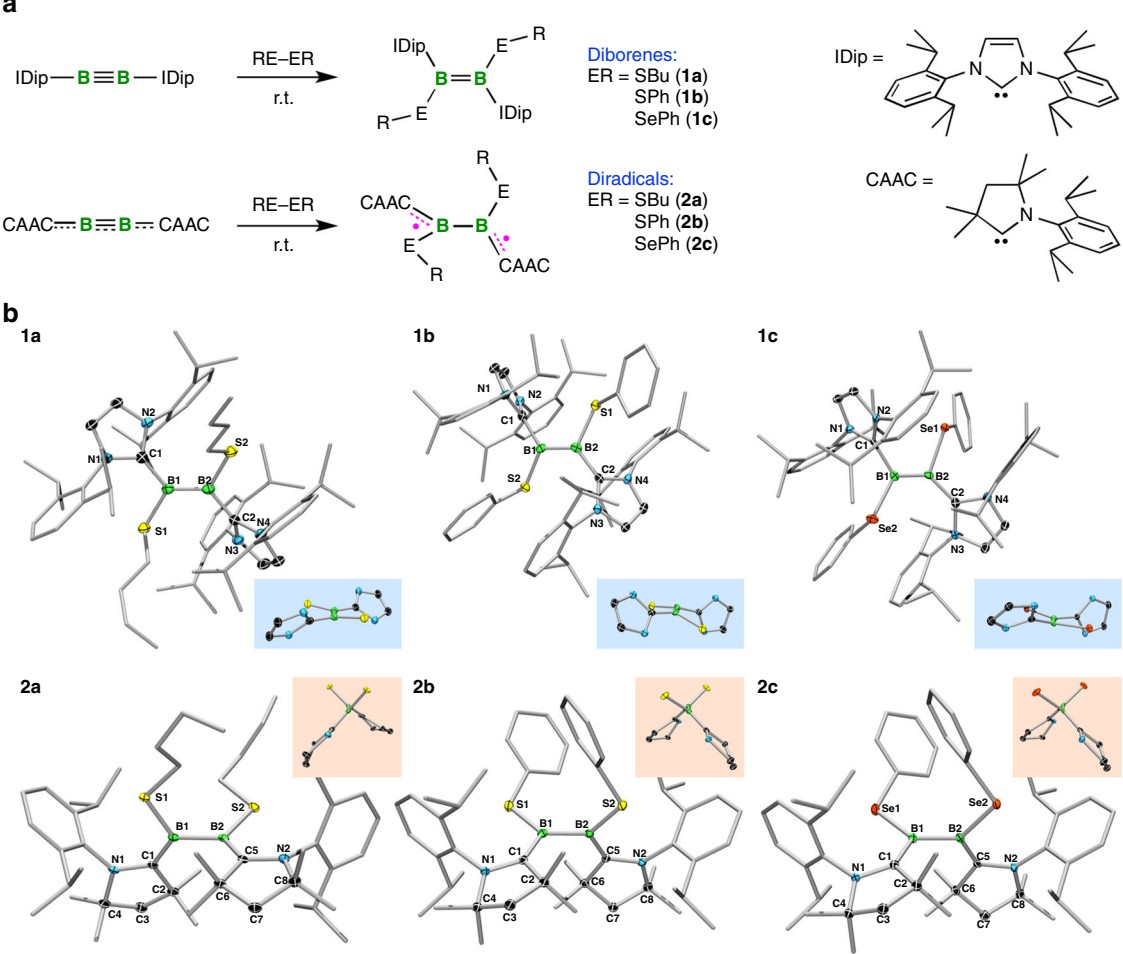

**Fig. 2** Synthesis and structures of diborenes **1** and diradicals **2**. **a** Synthesis of diborenes **1** and diradicals **2**. **b** Solid-state structures of diborenes **1** and diradicals **2**. Blue insets: views along the roughly planar B=B units of **1a**–**c**. Orange insets: views along the B–B axis showing the near-perpendicular boron planes

the midpoint of the molecules, the N-C-B-S/Se units are near-planar, with relatively small N-C-B-S/Se torsion angles (**2a**: 1.7 (2)°, 14.6(2)°; **2b**: 17.4(2)°, 1.5(2)°; **2c**: 18.7(3)°, 0.1(4)°) suggesting significant π-electron delocalization over this portion of the molecule. The B-S/Se distances of **2a**–**c** ($d$(B–S): 1.84–1.86 Å; $d$(B–Se): 1.99–2.00 Å) were found to be slightly shorter than those of **1a**–**c** ($d$(B–S): 1.88–1.92 Å; $d$(B–Se): 2.01–2.06 Å), and the B–C distances of **2a**–**c** (1.53–1.55 Å) are likewise shorter than those of **1a**–**c** (1.57–1.62 Å).

**EPR and magnetic measurements**. The significantly greater delocalization in the C-B-S/Se portion of **2a**–**c**, the apparent absence of B–B multiple bonding, and the inability to procure NMR data of the compounds suggested that they may be captodatively stabilized[30] triplet diradicals, each with two push–pull systems involving the π-donating S/Se atom and the π-accepting CAAC carbene carbon atom (i.e., LRB(•)–B(•)RL; Fig. 2a, bottom). Furthermore, the coplanar B and C atoms and short B–C bonds of **2a**–**c** are reminiscent of our related neutral monoradical species [DurB(Cl)(CAAC)]• reported in 2014[31]. In order to confirm the diradical nature of **2a**–**c**, we performed EPR spectroscopy on **2a**–**c** in frozen 2-methyltetrahydrofuran glass. While we observed an intense half-field transition for **2a** and **2b** (Fig. 3a, Supplementary Figs. 12–16), only a weak signal was observed for compound **2c** (Supplementary Fig. 17). The obtained spectra of **2a** and **2b** are dominated by rhombic zero-field splitting (ZFS)

due to dipole interaction, indicating that the spin system in question is a triplet state. Analysis of the spectra yields estimates of the ZFS parameters (**2a**: $D = 0.036$ cm$^{-1}$, $E = 0.0054$ cm$^{-1}$; **2b**: $D = 0.042$ cm$^{-1}$, $E = 0.0077$ cm$^{-1}$), the axial parts of which correspond to interspin distances of 4.5 and 4.4 Å (assuming point dipole approximation), respectively. These distances are significantly longer than the ca. 1.70–1.73 Å boron–boron distances found by X-ray crystallography, and match better the distance between the two CAAC carbene carbon atoms (**2a**: ca. 4.0 Å; **2b**: ca. 3.9 Å). Thus, the two unpaired electrons must be significantly delocalized into the substituents, most likely toward the carbene carbon atoms, which is in line with that expected by the captodative effect of the π-donor nitrogen and π-acceptor boron atoms.

To probe the exchange interaction between spin carriers in diradical **2a**, temperature-dependent magnetic measurements were performed on a solid sample (Fig. 3b). The effective magnetic moment increases from 2.50 μ$_B$ at RT to 2.81 μ$_B$ at 10 K, clearly indicating the presence of ferromagnetic interaction inside the molecule. Quantitative analysis based on the $\hat{H} = -2J\hat{S}_1\hat{S}_2$ spin Hamiltonian allowed determination of the exchange coupling parameter to be $J = +15$ cm$^{-1}$, i.e., the ground state of the molecule is a triplet and lies $2J = 30$ cm$^{-1}$ (0.086 kcal mol$^{-1}$) lower than the singlet state. The exchange coupling determined from solid-state SQUID measurements is also reflected to some degree in the temperature-dependence of the signal intensities observed in the solution-state EPR spectra of **2a** (Supplementary Fig. 14).

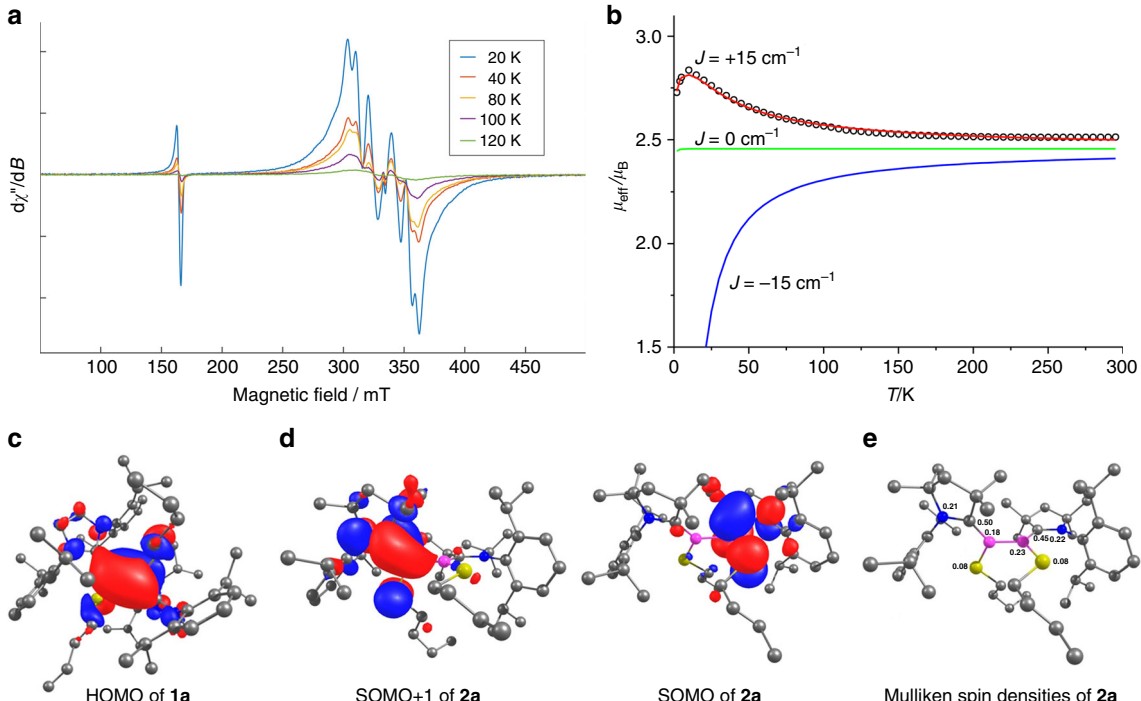

**Fig. 3** Spectroscopic, magnetic, and computational studies of compounds **1a** and **2a**. **a** Temperature-dependent EPR spectra of 1 mM **2a** in 2-Me-THF. Experimental parameters: temperature = 20–120 K; microwave frequency = 9.38 GHz; microwave power = 0.1 mW; modulation amplitude = 2 G; conversion time = 60 ms; modulation frequency = 100 kHz. **b** Temperature dependence of the magnetic moment of **2a**. Black cycles: experimental data. Red line: best fit data for the ferromagnetically coupled case with $J = +15$ cm$^{-1}$ and $g = 2.01$. For comparison, the theoretical data for the non-coupled ($J = 0$ cm$^{-1}$, green line) and antiferromagnetically coupled ($J = -15$ cm$^{-1}$, blue line) cases are also depicted. **c** HOMO plot of **1a** at the M05-2X/def2-SVP level. **d** SOMO plots of **2a** at the M05-2X/def2-SVP level. **e** Mulliken spin densities of relevant atoms of **2a** at the M05-2X/def2-SVP level

**Quantum-chemical investigations**. Density functional theory (DFT) calculations for diamagnetic diborenes **1a–c** and the diradicals **2a–c** (Supplementary Figs. 22–39) agree with the experimental findings that the singlet states of **1a–c** are always more stable than the triplet states, whereas the triplet states of **2a–c** are always more stable than the singlet states. The calculated bond lengths (Supplementary Figs. 22–39) and Wiberg bond indices (WBIs) support the proposed bonding arrangements depicted in Fig. 2a, with B=B double bond character in **1a–c** (**1a**: 1.55, **1b**: 1.58, **1c**: 1.60) and single B–B bond character in **2a–c** (**2a**: 0.97, **2b**: 0.98, **2c**: 0.99). Figure 3c, d (and Supplementary Figs. 40–45) illustrate the stark contrasts between the highest-energy occupied orbitals of diborenes **1a–c** and diradicals **2a–c**. The former, as exemplified by **1a**, possess highest occupied molecular orbitals (HOMOs; Fig. 3c) consisting of bonding B–B π interactions that extend slightly toward the $p_\pi$ orbital of the carbene carbon atoms. The latter, as exemplified by **2a**, have two near-degenerate singly-occupied molecular orbitals (SOMOs; Fig. 3d), each of which is delocalized over one N-C-B-S/Se moiety, with a bonding interaction between B and C, weak antibonding between C and N, and weaker antibonding between B and S/Se. These singly-occupied orbitals clearly illustrate the presence of two radical centers in each of **2a-c**, and that each unpaired electron resides in one respective N-C-B-S/Se π system. The calculated Mulliken spin densities of **2a** (Fig. 3e) show that the unpaired electrons reside predominantly in the CAAC carbon $p_\pi$ orbitals (0.50 and 0.45 e), with lower but roughly equivalent spin density on the B (0.18 and 0.23 e) and N (0.21 and 0.22 e) atoms.

While DFT computations and experiment agree in terms of geometry and the energetic order of the singlet and triplet states, the DFT computations considerably overestimate the size of the singlet–triplet gap (S–T gap) of compound **2a**. The

overestimation may result from the need for multi-reference approaches[37,38] to describe the singlet and triplet states on equal footing. Because **2a** is too large for such costly approaches, we investigated the error of the DFT approaches for a smaller model system (**2a′**; Supplementary Figure 46), which reflects all of the essential binding effects of **2a** but is sufficiently small for multi-reference calculations. The results (Supplementary Table 3) show that all functionals indeed overestimate the stability of the triplet state considerably. While the very accurate NEVPT2[39,40]/def2-TZVP approach predicts an S–T gap of only −0.4 kcal mol$^{-1}$ (triplet lower than singlet), the values obtained with DFT range from −4 to −28 kcal mol$^{-1}$. The MN12-L/6–311G(d,p) approach used to estimate the S–T gap of the full system **2a** overestimates the stability of the triplet state by about 6–7 kcal mol$^{-1}$. Transferring this error to the predictions of the MN12L[41]/6–311G(d,p)[42,43] approach for the full system **2a** (−6.4 kcal mol$^{-1}$), a near-zero S–T gap results, which is in line with the measured value.

**Electrochemical investigations**. The electrochemical behavior of **2a–c** also distinguishes these diradicals from diborenes **1a–c** (Supplementary Figs. 20 and 21). Diborenes **1a-c** show facile oxidation events in their cyclic voltammograms (with no apparent reduction processes), as previously observed for related diborenes[44]. In contrast, diradicals **2a–c** show less-facile oxidation events, with additional reduction waves, in line with the presence of the more π-accepting CAAC ligands. It should also be noted that variable-temperature NMR experiments suggest that the diamagnetic diborenes **1** and the diradical species **2** do not convert to their alternative conformations upon heating to 80 °C.

**Conclusion**. The synthesis of diborenes **1a–c** and their twisted congeners **2a–c** marks the first isolation and structural authentication of diamagnetic, multiply-bound molecules and their diradical, single-bond congeners, products of formal homolysis of the central multiple bond. Crystallographic, EPR spectroscopic, magnetic and theoretical studies confirm that **2a–c** are diradical species. Spectroscopic and computational studies also confirm the substantial energy differences between the singlet and triplet states of the molecules, with the singlet states being favored in **1a–c** and the triplet states in **2a–c**.

## Methods

**Synthetic methods**. All reactions were carried out under an atmosphere of dry argon using standard Schlenk line and glovebox techniques. NMR spectra (Supplementary Figures 1–11) were obtained from a Bruker Avance 500 NMR spectrometer ($^1$H: 500.13 MHz, $^{11}$B: 160.46 MHz, $^{13}$C{$^1$H}: 125.76 MHz) at RT. Chemical shifts ($\delta$) are given in ppm and internally referenced to the carbon nuclei ($^{13}$C{$^1$H}) or residual protons ($^1$H) of the solvent. $^{11}$B NMR spectra were referenced to the external standard [BF$_3$·OEt$_2$]. Microanalyses (C, H, N) were performed on an Elementar vario MICRO cube elemental analyzer. Melting points were determined with a Mettler Toledo 823e DSC in sealed ampules with a ramp rate of 10 °C min$^{-1}$. UV/vis absorption spectra (Supplementary Figures 18 and 19) were recorded on a JASCO V-660 UV/vis spectrometer. Cyclic voltammetry experiments were performed using a Gamry Instruments Reference 600 potentiostat. Deuterated benzene was purchased from Sigma Aldrich and dried over molecular sieves. Other solvents were dried by storage over, and distillation from, potassium benzophenone (THF, diethyl ether), Na/K alloy (pentane, hexane), or sodium (toluene, benzene) under an argon atmosphere. Solvents were stored under argon over activated 4 Å molecular sieves. Diboryne precursors B$_2$IDip$_2$[24] and B$_2$CAAC$_2$[36] were prepared according to literature methods. Dimethyl disulfide, dibutyl disulfide, diphenyl disulfide, and diphenyl diselenide were purchased from commercial sources.

*Synthesis of B$_2$(IDip)$_2$(SBu)$_2$, (1a)*: Dibutyldisulfide (34.7 µL, 188 µmol) was added dropwise to a stirred solution of B$_2$(IDip)$_2$ (100 mg, 125 µmol) in pentane (10 mL). The mixture was stirred for 18 h at RT and all volatiles were removed in vacuo. The dark green residue was dried in vacuo at 80 °C for 2 h to remove the excess dibutyldisulfide, extracted with 5 mL pentane, and crystallized from a slowly evaporating hexane/ether solution. The crystalline material was washed with pentane (2 × 1 mL) yielding 73.4 mg (75.1 µmol, 60%) of pure **1a** as a green solid. $^1$H NMR (500.1 MHz, toluene-$d_8$): $\delta = 7.19$–7.14 (m, 4 H, CH$_{Aryl}$), 7.04 (s, 4 H, CH$_{Aryl}$), 7.02 (s, 4 H, CH$_{Aryl}$), 6.36 (s, 4 H, CH$_{NHC}$), 3.02 (br, 8 H, CH$_{iPr}$), 1.74 (t, $^3J_{H,H} = 7.78$ Hz, 4 H, SBu-CH$_2$), 1.23 (d, $^3J_{H,H} = 6.1$ Hz, 24 H, iPr-CH$_3$), 1.17–1.05 (m, 8 H, SBu-CH$_3$), 0.98 (d, $^3J_{H,H} = 6.8$ Hz, 24 H, iPr-CH$_3$), 0.88 (t, $^3J_{H,H} = 7.78$ Hz, 6 H, SBu-CH$_3$). $^{13}$C NMR (125.8 MHz, toluene-$d_8$) $\delta = 176.8$ (C$_{NHC}$), 147.0 (C$_q$), 139.2 (C$_q$), 129.1 (CH$_{Aryl}$), 124.4 (CH$_{Aryl}$), 123.8 (CH$_{NHC}$), 43.1 (SBu-CH$_2$), 34.9 (SBu-CH$_2$), 29.1 (CH$_{iPr}$), 26.2 (iPr-CH$_3$), 24.3 (iPr-CH$_3$), 24.0 (SBu-CH$_2$), 14.7 (SBu-CH$_3$). $^{11}$B NMR (160.5 MHz, toluene-$d_8$): $\delta = 30$ (s) ppm. Elemental analysis calculated for C$_{62}$H$_{90}$B$_2$N$_4$S$_2$[+hexane]: C 76.81%, H 9.86%, N 5.27%, S 6.03%; found: C 76.86%, H 9.63%, N 5.62%, S 5.82%.

*Synthesis of B$_2$(IDip)$_2$(SPh)$_2$, (1b)*: B$_2$IDip$_2$ (50.0 mg, 62.6 µmol) and diphenyldisulfide (13.7 mg, 62.7 µmol) were dissolved in benzene (10 mL) and stirred for 14 h. The solvent was evaporated and the residue was extracted with pentane (2 × 5 mL). The pentane phase was removed under vacuum giving pure **1b** (16.8 mg, 16.5 µmol, 26%) as a purple solid. Suitable crystals for X-ray diffraction were obtained by slow evaporation of a pentane solution. Decomp.: 221 °C. $^1$H NMR (500.1 MHz, C$_6$D$_6$): $\delta = 7.04$ (t, $^3J_{H,H} = 7.6$ Hz, 4 H, CH$_{Aryl}$), 6.91 (d, $^3J_{H,H} = 7.6$ Hz, 8 H, CH$_{Aryl}$), 6.85 (br, 4 H, CH$_{Aryl}$), 6.79 (br, 6 H, CH$_{Aryl}$), 6.29 (s, 4 H, CH$_{NHC}$), 3.20 (br, 8 H, CH$_{iPr}$), 1.05 (m, 24 H, iPr-CH$_3$), 0.97 (d, $^3J_{H,H} = 6.8$ Hz, 24 H, iPr-CH$_3$) ppm. $^{13}$C NMR (125.8 MHz, C$_6$D$_6$) $\delta = 174.7$ (C$_q$), 148.8 (C$_q$), 146.3 (C$_q$), 137.6 (C$_q$), 129.3 (CH$_{Aryl}$), 128.9 (CH$_{Aryl}$), 127.7 (CH$_{Aryl}$), 124.2 (CH$_{Aryl}$), 124.0 (CH$_{NHC}$), 120.1 (CH$_{Aryl}$), 28.9 (CH$_{iPr}$), 26.6 (iPr-CH$_3$), 22.7 (iPr-CH$_3$) ppm. $^{11}$B NMR (160.5 MHz, C$_6$D$_6$): $\delta = 25$ (s) ppm. Elemental analysis calculated for C$_{66}$H$_{82}$B$_2$N$_4$S$_2$: C 77.94%, H 8.13%, N 5.51%, S 6.30%; found: C 78.12%, H 8.26%, N 5.58%, S 6.25%.

*Synthesis of B$_2$(IDip)$_2$(SePh)$_2$, (1c)*: A solution of B$_2$IDip$_2$ (50.0 mg, 62.6 µmol) and 1,2-diphenyldiselenide (19.5 mg, 62.6 µmol) in benzene (10 mL) was stirred for 14 h. After the solvent was removed, the residue was extracted with pentane (2 × 5 mL). The solvent was evaporated under vacuum giving pure **1c** (15.8 mg, 14.2 µmol, 23%) as a purple solid. Suitable crystals for X-ray diffraction were obtained by slow evaporation of a pentane solution. Decomp.: 128 °C. $^1$H NMR (500.1 MHz, C$_6$D$_6$): $\delta = 7.05$ (t, $^3J_{H,H} = 7.6$ Hz, 4 H, CH$_{Aryl}$), 6.96 (d, $^3J_{H,H} = 7.4$ Hz, 4 H, CH$_{Aryl}$), 6.93 (d, $^3J_{H,H} = 7.6$ Hz, 8 H, CH$_{Aryl}$), 6.88 (t, $^3J_{H,H} = 7.4$ Hz, 2 H, CH$_{Aryl}$), 6.76 (t, $^3J_{H,H} = 7.4$ Hz, 4 H, CH$_{Aryl}$), 6.33 (s, 4 H, CH$_{NHC}$), 3.29 (br, 8 H, CH$_{iPr}$), 1.02 (m, 24 H, iPr-CH$_3$), 0.98 (d, $^3J_{H,H} = 6.8$ Hz, 24 H, iPr-CH$_3$) ppm. $^{13}$C NMR (125.8 MHz, C$_6$D$_6$) $\delta = 174.7$ (C$_q$), 146.5 (C$_q$), 143.2 (C$_q$), 137.53 (C$_q$), 137.52 (C$_q$), 131.3 (CH$_{Aryl}$), 129.5 (CH$_{Aryl}$), 127.7 (CH$_{Aryl}$), 124.3 (CH$_{Aryl}$), 124.1 (CH$_{NHC}$), 121.3 (CH$_{Aryl}$), 29.0 (CH$_{iPr}$), 26.7 (iPr-CH$_3$), 22.8 (iPr-CH$_3$) ppm. $^{11}$B NMR (160.5 MHz, C$_6$D$_6$): $\delta = 22$ (s) ppm. $^{77}$Se NMR (95.4 MHz, C$_6$D$_6$): $\delta = 175.6$ ppm. Elemental

analysis calculated for C$_{66}$H$_{82}$B$_2$N$_4$Se$_2$: C 71.36%, H 7.44%, N 5.04%; found: C 70.68%, H 7.66%, N 4.66%.

*Synthesis of B$_2$(CAAC)$_2$(SBu)$_2$, (2a)*: Dibutyl disulfide (141 µL, 743 µmol) was added dropwise to a stirred solution of B$_2$(CAAC)$_2$ (400 mg, 675 µmol) in pentane (20 mL) at −40 °C. The mixture was stirred for 24 h at RT and all volatiles were removed in vacuo. The dark yellow solid was dried in vacuo, dissolved in hexane (10 mL), and black crystals were obtained by slow evaporation of the solvent. The crystalline material was washed with pentane (2 × 1 mL) yielding 291 mg (379 µmol, 56%) of pure **2a** as a black solid. Elemental analysis calculated for C$_{48}$H$_{80}$B$_2$N$_2$S$_2$: C 74.78%, H 10.46%, N 3.63%, S 8.32%; found: C 75.15%, H 10.37%, N 3.70%, S 8.01%.

*Synthesis of B$_2$(CAAC)$_2$(SPh)$_2$, (2b)*: Pentane (20 mL at −40 °C) was added to a mixture of B$_2$(CAAC)$_2$ (100 mg, 168 µmol) and diphenyl disulfide (36.8 mg, 168 µmol). The mixture was stirred for 3 h at RT, filtered and the residual solid was washed with pentane (3 × 5 mL). The brown solid was dried in vacuo, dissolved in benzene (10 mL), and black crystals were obtained by slow evaporation of the solvent. The crystalline material was washed with pentane (2 × 1 mL) yielding 53.4 mg (65.9 µmol, 39%) of pure **2b** as a black solid. Decomp.: 138 °C. Elemental analysis calculated for C$_{52}$H$_{72}$B$_2$N$_2$S$_2$: C 77.02%, H 8.95%, N 3.45%, S 7.91%; found: C 77.57%, H 8.97%, N 3.36%, S 7.58%.

*Synthesis of B$_2$(CAAC)$_2$(SePh)$_2$, (2c)*: Pentane (20 mL at −40 °C) was added to a mixture of B$_2$(CAAC)$_2$ (100 mg, 168 µmol) and diphenyl diselenide (52.7 mg, 168 µmol). The mixture was stirred for 3 h at RT, filtered and the residual solid was washed with pentane (2 × 5 mL). The brown solid was dried in vacuo, dissolved in benzene (10 mL), and black crystals were obtained by slow evaporation of the solvent. The crystalline material was washed with pentane (2 × 1 mL) yielding 36.2 mg (40.0 µmol, 24%) of pure **2c** as a black solid. Decomp.: 191 °C. Elemental analysis calculated for C$_{52}$H$_{72}$B$_2$N$_2$Se$_2$: C 69.04%, H 8.02%, N 3.10%; found: C 69.24%, H 7.95%, N 2.74%.

**EPR spectroscopic methods**. EPR measurements at the X-band (9.4 GHz) were carried out using a Bruker ELEXSYS E580 CW EPR spectrometer equipped with an Oxford Instruments helium cryostat (ESR900) and a MercuryiTC temperature controller. The spectral simulations were performed using MATLAB 8.6.0.267246 (R2015b) and the EasySpin 5.1.9 toolbox[45]. The EPR spectra of **2a** and **2b** show an intense half-field signal but only partially resolved zero-field splittings with no visible hyperfine structure. Attempts to improve the resolution by lowering the concentration or changing the solvent were not successful. For **2c**, no clear sign of a triplet ground state could be obtained.

**Electrochemical methods**. Cyclic voltammetry experiments were performed using a Gamry Instruments Reference 600 potentiostat. A standard three-electrode cell configuration was employed using a platinum disk working electrode, a platinum wire counter electrode, and a silver wire, separated by a *Vycor* tip, serving as the reference electrode. Formal redox potentials are referenced to the ferrocene/ferrocenium ([Cp$_2$Fe]$^{+/0}$) redox couple by using ferrocene as an internal standard. Tetrabutylammonium hexafluorophosphate ([Bu$_4$N][PF$_6$]) was employed as the supporting electrolyte. Compensation for resistive losses (iR drop) was employed for all measurements.

**Crystallographic methods**. The crystal data of B$_2$IDip$_2$(SBu)$_2$ (**1a**), B$_2$IDip$_2$(SPh)$_2$ (**1b**), B$_2$IDip$_2$(SePh)$_2$ (**1c**), and B$_2$CAAC$_2$(SBu)$_2$ (**2a**) were collected on a Bruker X8-APEX II diffractometer with a CCD area detector and multi-layer mirror-mono-chromated Mo$_{K\alpha}$ radiation. The structure was solved using the intrinsic phasing method (SHELXT)[46], refined with the SHELXL program[47], and expanded using Fourier techniques. All non-hydrogen atoms were refined anisotropically. Hydrogen atoms were included in structure factor calculations. All hydrogen atoms were assigned to idealized geometric positions.

*Crystal data for B$_2$IDip$_2$(SBu)$_2$ (1a)*: The displacement parameters of atoms C63–C74 were restrained to the same value with similarity restraint SIMU. C$_{68}$H$_{104}$B$_2$N$_4$S$_2$, $M_r = 1063.29$, green block, 0.294 × 0.257 × 0.125 mm$^3$, monoclinic space group $P2_1/c$, $a = 14.423(5)$ Å, $b = 12.522(5)$ Å, $c = 36.640(17)$ Å, $\beta = 96.235(3)°$, $V = 6578(5)$ Å$^3$, $Z = 4$, $\rho_{calcd} = 1.074$ g cm$^{-3}$, $\mu = 0.122$ mm$^{-1}$, $F(000) = 2328$, $T = 103$ (2) K, $R_1 = 0.0980$, $wR^2 = 0.1550$, 13420 independent reflections [$2\theta \leq 52.744°$], and 768 parameters. CCDC-1567701.

*Crystal data for B$_2$IDip$_2$(SPh)$_2$ (1b)*: C$_{66}$H$_{82}$B$_2$N$_4$S$_2$, $M_r = 1017.09$, red block, 0.25 × 0.14 × 0.05 mm$^3$, monoclinic space group $P2_1/c$, $a = 22.2980(16)$ Å, $b = 24.4556(17)$ Å, $c = 42.972(3)$ Å, $\beta = 91.133(2)°$, $V = 23428(3)$ Å$^3$, $Z = 16$, $\rho_{calcd} = 1.153$ g cm$^{-3}$, $\mu = 0.134$ mm$^{-1}$, $F(000) = 8768$, $T = 100(2)$ K, $R_1 = 0.1465$, $wR^2 = 0.1925$, 48137 independent reflections [$2\theta \leq 52.746°$], and 2730 parameters. CCDC-1567705.

*Crystal data for B$_2$IDip$_2$(SePh)$_2$ (1c)*: C$_{66}$H$_{82}$B$_2$N$_4$Se$_2$, $M_r = 1110.89$, red block, 0.35 × 0.15 × 0.08 mm$^3$, monoclinic space group $P2_1/c$, $a = 22.4904(11)$ Å, $b = 24.4964(13)$ Å, $c = 43.042(2)$ Å, $\beta = 91.335(2)°$, $V = 23707(2)$ Å$^3$, $Z = 16$, $\rho_{calcd} = 1.245$ g cm$^{-3}$, $\mu = 1.291$ mm$^{-1}$, $F(000) = 9344$, $T = 100(2)$ K, $R_1 = 0.0804$, $wR^2 = 0.0917$, 48482 independent reflections [$2\theta \leq 52.744°$], and 2729 parameters. CCDC-1567706.

*Crystal data for B$_2$CAAC$_2$(SBu)$_2$ (2a)*: C$_{51}$H$_{87}$B$_2$N$_2$S$_2$, $M_r = 813.96$, orange block, 0.335 × 0.257 × 0.112 mm$^3$, triclinic space group $P\bar{1}$, $a = 10.383(5)$ Å, $b = 14.048(9)$ Å, $c = 17.857(8)$ Å, $\alpha = 75.299(15)°$, $\beta = 84.42(3)°$, $\gamma = 77.545(19)°$, $V = 2458(2)$ Å$^3$,

$Z = 2$, $\rho_{calcd} = 1.100$ g cm$^{-3}$, $\mu = 0.143$ mm$^{-1}$, $F(000) = 898$, $T = 100(2)$ K, $R_1 = 0.0532$, $wR^2 = 0.1087$, 10053 independent reflections [$2\theta \leq 52.744°$], and 533 parameters. CCDC-1567703.

The crystal data of B$_2$CAAC$_2$(SPh)$_2$ (**2b**) and B$_2$CAAC$_2$(SePh)$_2$ (**2c**) were collected on a Bruker D8 Quest diffractometer with a CMOS area detector and multi-layer mirror-monochromated Mo$_{K\alpha}$ radiation. The structure was solved using the intrinsic phasing method (ShelXT)[46], refined with the ShelXLprogram[47], and expanded using Fourier techniques. All non-hydrogen atoms were refined anisotropically. Hydrogen atoms were included in structure factor calculations. All hydrogen atoms were assigned to idealized geometric positions.

*Crystal data for B$_2$CAAC$_2$(SPh)$_2$ (**2b**)*: C$_{52}$H$_{72}$B$_2$N$_2$S$_2$, $M_r = 810.85$, orange block, $0.173 \times 0.161 \times 0.122$ mm$^3$, monoclinic space group P2$_1$/n, $a = 9.9798(15)$ Å, $b = 18.647(4)$ Å, $c = 24.919(5)$ Å, $\beta = 97.909(15)°$, $V = 4593.2(15)$ Å$^3$, $Z = 4$, $\rho_{calcd} = 1.173$ g cm$^{-3}$, $\mu = 0.153$ mm$^{-1}$, $F(000) = 1760$, $T = 100(2)$ K, $R_1 = 0.0422$, $wR^2 = 0.0961$, 9368 independent reflections [$2\theta \leq 52.746°$], and 539 parameters. CCDC-1567702.

*Crystal data for B$_2$CAAC$_2$(SePh)$_2$ (**2c**)*: C$_{52}$H$_{72}$B$_2$N$_2$Se$_2$, $M_r = 904.65$, orange block, $0.061 \times 0.055 \times 0.025$ mm$^3$, monoclinic space group P2$_1$/n, $a = 10.060(7)$ Å, $b = 18.707(7)$ Å, $c = 24.857(14)$ Å, $\beta = 98.86(3)°$, $V = 4622(5)$ Å$^3$, $Z = 4$, $\rho_{calcd} = 1.300$ g cm$^{-3}$, $\mu = 1.637$ mm$^{-1}$, $F(000) = 1904$, $T = 100(2)$ K, $R_1 = 0.0628$, $wR^2 = 0.0784$, 9321 independent reflections [$2\theta \leq 52.746°$], and 539 parameters. CCDC-1567704.

**Computational methods**. Geometry optimizations of all complexes were carried out at the M05-2X[48], B3LYP[49–51], and BP86[50,52] levels in conjunction with the def2-SVP[53] basis set. The structures were characterized as minima by frequency calculation. The NBO[54] analyses were carried out at the M05-2X/def2-SVP level (Supplementary Table 2). The calculations were performed with the Gaussian 09, Revision D.01 program package[55]. Density functional theory (DFT) calculations were carried out on diamagnetic diborenes **1a–c** and the diradicals **2a–c** in their singlet and triplet states (Supplementary Figs. 22–39). Calculations using the functionals M05-2X, B3LYP, and BP86 show that the singlet states of **1a–c** are always more stable than the triplet states ($\Delta E_{S-T}$ of **1a**: 16.7, 17.8, 19.1, **1b**: 24.4, 22.2, 21.8, **1c**: 26.3, 23.6, 23.0 kcal mol$^{-1}$ at the M05-2X, B3LYP, and BP86 levels, respectively) whereas the triplet states of **2a-c** are always more stable than the singlet states ($\Delta E_{S-T}$ of **2a**: −33.2, −28.6, −23.6; **2b**: −23.6, −20.3, −15.2; **2c**: −28.5, −20.6, −20.1 kcal mol$^{-1}$ at the M05-2X, B3LYP, and BP86 levels, respectively) (Supplementary Table 1). The bond lengths calculated at the M05-2X/def2-SVP level (Supplementary Figs. 22–27) are in better agreement with the experimental bond lengths than those calculated at the B3LYP/def2-SVP (Supplementary Figs. 28–33) and BP86/def2-SVP levels (Supplementary Figs. 34–39); our discussions thus focus on calculations performed at the former level.

*High level computations for model compound **2a′***: For **2a**, measurements indicate that the triplet state is only slightly more stable than the corresponding singlet state. In contrast, DFT computations employing various functionals predict that the triplet state lies more than 20 kcal mol$^{-1}$ below the corresponding singlet state. The large overestimation of the S–T gap may partially result from solid-state effects; however, the DFT functionals used may also underestimate static correlation effects, which stabilize the biradical singlet state of **2a** with respect to the corresponding triplet ground states. More insights into the reasons for the differences between measured and computed values can only be obtained from high-level multi-reference computations; however, compound **2a** is too large for such approaches.

To obtain an estimate of the inaccuracies of DFT approaches, we first used the MN12L[41]/6-311G(d,p)[42,43] approach to optimize the geometry of the lowest-lying singlet and triplet states of **2a**. Both optimizations were started from the X-ray structure of the triplet ground state. The computed singlet geometry does not represent the global minimum of the singlet potential energy surface, which possesses a nearly planar C$^{CAAC}$-B-B-C$^{CAAC}$ orientation. However, only the local minimum lying near the triplet equilibrium structure will not be populated in the measurements because the global minimum of the singlet state is separated from the triplet equilibrium geometry by high-energy barriers. For the computed geometries (global minimum of triplet and local minimum of singlet state), the MN12L/6-311G(d,p) approach predicts an S–T gap of about −6.4 kcal mol$^{-1}$ (triplet below singlet state).

To enable high-level multi-reference approaches, we then created a closely related model system **2a′** in which we replaced the bulky substituents as indicated in Supplementary Figure 46. For this model compound, we reoptimized the carbon-heteroatom and the C–H distances of the new substituents but retained the overall geometries of the **2a** systems to include the influence of steric effects on the geometry. The model system **2a′** contains all essential binding ingredients of **2a** but is sufficiently small to perform high-level NEVPT2[39,40]/def2-TZVP benchmark calculations for the S–T gap. To compare the predicted S–T gaps, we performed single-point calculations employing the geometries described above.

The results are summarized in Supplementary Table 3. It shows that all functionals overestimate the stability of the triplet state in comparison to the better-suited multi-reference NEVPT2 approach. The errors range from about −5 kcal mol$^{-1}$ to nearly −29 kcal mol$^{-1}$, showing that such DFT calculations have to be handled with care.

Cartesian coordinates for all of the calculated compounds are given in Supplementary Table 4.

**Magnetic methods**. Temperature-dependent magnetic susceptibility measurements for **2a** were carried out with a Quantum-Design MPMS-XL-5 SQUID magnetometer equipped with a 5-Tesla magnet in the range from 295 to 2.0 K at a magnetic field of 0.5 T. The powdered sample was contained in a Teflon bucket and fixed in a non-magnetic sample holder. Each raw data file for the measured magnetic moment was corrected for the diamagnetic contribution of the Teflon bucket according to $M^{dia}$(bucket) $= \chi_g \times m \times H$, with an experimentally obtained gram susceptibility of the Teflon bucket. The molar susceptibility data were corrected for the diamagnetic contribution using Pascal's constants[56]. Experimental data were modeled with the *julX* program[57] using a fitting procedure to the spin Hamiltonian $\hat{H} = -2J\hat{S}_1\hat{S}_2 + g\mu_B\vec{B}(\vec{S}_1 + \vec{S}_2)$. A slightly better fit quality was obtained using an additional diamagnetic contribution (−140×10$^{-6}$ cm$^3$ mol$^{-1}$) and some weak intermolecular interactions (−0.12 K). The latter were considered in a mean field approach by using a Weiss temperature $\Theta$[58]. The Weiss temperature $\Theta$ (defined as $\Theta = zJ_{inter}S(S + 1)/3k$) relates to intermolecular interactions $zJ_{inter}$, where $J_{inter}$ is the interaction parameter between two nearest neighbor magnetic centers, $k$ is the Boltzmann constant (0.695 cm$^{-1}$ K$^{-1}$), and $z$ is the number of nearest neighbors.

**Data availability**. The data that support the findings of this study are available from the corresponding author on reasonable request. The X-ray crystallographic coordinates for structures reported in this study have been deposited at the Cambridge Crystallographic Data Centre (CCDC), under deposition numbers 1567701−1567706. These data can be obtained free of charge from The Cambridge Crystallographic Data Centre via http://www.ccdc.cam.ac.uk/data_request/cif.

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

## Acknowledgements

Partial financial support for this project was provided by the European Research Council (ERC) under the European Union Horizon 2020 Research and Innovation Program (Boron-Boron Multiple Bonding, Advanced Grant agreement no. 669054, to H.B.), and the Deutsche Forschungsgemeinschaft Research Training Group GRK2112, Molecular Biradicals: Structure, Properties and Reactivity (H.B., B.E., and R.M.).

## Author contributions

H.B. conceived the study. J.B., T.D., W.C.E., K.H., and M.H. performed the synthetic experiments. M.A.C., E.W., M.R., R.M., and B.E. performed the computational studies. I. K. and E.B. acquired and analyzed the EPR data. S.D. and F.M. performed the magnetic measurements. M.A.C., R.D.D., and W.C.E. prepared the manuscript. All authors read and commented on the manuscript.

## Additional information

**Competing interests:** The authors declare no competing interests.

