## [Peer Review File(PDF 139 kb) · Nature Communications]

REVIEWERS' COMMENTS:

Reviewer #1 (Remarks to the Author):

The manuscript by Braunschweig and coworkers describes the isolation, characterization and structure determination of the first biradical which results from a 90 degrees twisting around a E=E double bond. The particular molecule which the authors studied is a B=B bond coordinated to two carbenes. The manuscript reports a really spectacular achievement, in particular the fact that an X-ray structure of the diradical is reported.

The authors support their claim, in addition to the X-ray structure by detailed EPR studies in a frozen matrix, electrochemical measurements and quantum mechanical calculations. The Supplementary Information is detailed, reliable and contains all the relevant information. In summary, this manuscript describes excellent science of great fundamental importance and is well written. It will be of interest to a wide audience of chemists and I strongly recommend its publication in Nature Communications.

Several minor comments:

1. The use of the words "products" in the title is somewhat misleading as chemists usually link the word "products" to "reaction products". Therefore, I suggest that the authors consider rephrasing the title.
2. It is interesting to know the rotation barrier in 1a and 2a (or their simpler models). Did the authors calculate them? If so, these calculations should be added to the manuscript.
3. In p.3 line 62: add ref. 22 after "experimentally".
4. P. 11 line 185: it is not clear what exactly the model system 2a is –it should be defined and it should be named 2a'.

Reviewer #2 (Remarks to the Author):

In the current manuscript Braunschweig and co-workers examine isomeric forms of a diboron system – one containing a formal B=B double bond, the other a diradical formulation derived from rotation about the B-B vector. The idea of twisting double bonds in this way is not new – as the authors acknowledge in their introduction – and has even been extended eg by Sekiguchi - to elements beyond carbon. However, the work contained in this manuscript is of significant impact because it shows how – by careful design of the ancillary substituents – a single system can be manipulated so that it adopts one conformationally isomeric form rather than the other. As such, I think this work is of the level of fundamental impact that I would expect from a publication in Nature Communications and I would recommend acceptance.

The strategy which the authors employ to discriminate between doubly bonded and diradical forms is by changing the electronic properties of the ancillary carbene donor – and in particular by using the cAAC family of ligands which are well known to offer strong pi acceptor properties, and which have been used in a number of previous systems to stabilize adjacent radical species. The resulting NHC and cAAC systems have been characterized very thoroughly – both computationally and experimentally - and the key magnetic properties of the diradical species have been examined in depth. As such, I have no significant comments to make concerning the scientific narrative of the paper, which reads very well.

The one minor omission – if one were being picky – would be some demonstration of the diradical character of 2 via onward reaction chemistry. This would be of particular interest given – for example – the role thought to be played by diradical forms in the chemistry of some of Power's dimetallyne systems. However, I would not regard this as being an essential addition for publication.

Two very minor points:

The description of the industrial use of olefins as being 'universal' is perhaps overstating the case!

Perhaps 'widespread' or if greater impact is desired - 'critical' but 'universal' implies that ALL industries use olefins....

The term 'captodatively' – is this the correct use of the term? Viehe's Acc Chem Res paper (ref 30) describes this as being the stabilization of radicals through the combined effect of donor and acceptor components on the radical centre (e.g. using alpha amino and cyano functions). The use of cAACs in stabilizing radical systems seems to be derived ostensibly from their pi acceptor characteristics and (as here) the consequent delocalization of spin density around the heterocycle.

Reviewer #3 (Remarks to the Author):

The authors synthesized and isolated the novel series of diborene compounds and their twisted congeners. The twisted congeners adopt diradical state, which are stabilized by the captodative effect of n donating chalcogen atoms and n -accepting carbene carbon atoms. Their structures and spin states were clearly revealed by experimental and theoretical studies including X-ray, NMR, EPR, SQUID, and DFT studies. The unique properties of the twisted diradical species were well discussed by comparison with their diborene congeners. These discussions are crucial for understanding of relationships between bonding nature and structures of compounds containing main group elements and will attract broad interest in basic chemistry and physical chemistry. This paper is therefore suitable for Nature Communications without changes.

REVIEWERS' COMMENTS:

Reviewer #1 (Remarks to the Author):

The manuscript by Braunschweig and coworkers describes the isolation, characterization and structure determination of the first biradical which results from a 90 degrees twisting around a E=E double bond. The particular molecule which the authors studied is a B=B bond coordinated to two carbenes. The manuscript reports a really spectacular achievement, in particular the fact that that an X-ray structure of the diradical is reported.

The authors support their claim, in addition to the X-ray structure by detailed EPR studies in a frozen matrix, electrochemical measurements and quantum mechanical calculations. The Supplementary Information is detailed, reliable and contains all the relevant information.

In summary, this manuscript describes excellent science of great fundamental importance and is well written. It will be of interest to a wide audience of chemists and I strongly recommend its publication in Nature Communications.

Several minor comments:

1. The use of the words “products” in the title is somewhat misleading as chemists usually link the word “products” to “reaction products”. Therefore, I suggest that the authors consider rephrasing the title.

RESPONSE: We have now altered the title according to the referee (and editorial) requests.

2. It is interesting to know the rotation barrier in 1a and 2a (or their simpler models). Did the authors calculate them? If so, these calculations should be added to the manuscript.

RESPONSE: Rotation barriers were not calculated due to the very large computational cost of performing this for such large molecules.

3. In p.3 line 62: add ref. 22 after “experimentally”.

RESPONSE: This has been done.

4. P. 11 line 185: it is not clear what exactly the model system 2a is –it should be defined and it should be named 2a’.

RESPONSE: This has been done, and a cross-reference to Supp. Fig. 46 has been added.

Reviewer #2 (Remarks to the Author):

In the current manuscript Braunschweig and co-workers examine isomeric forms of a diboron system – one containing a formal B=B double bond, the other a diradical formulation derived from rotation about the B-B vector. The idea of twisting double bonds in this way is not new – as the authors acknowledge in their introduction – and has even been extended eg by Sekiguchi - to elements beyond carbon. However, the work contained in this manuscript is of significant impact because it shows how – by careful design of the ancillary substituents – a single system can be manipulated so that it adopts one conformationally

isomeric form rather than the other. As such, I think this work is of the level of fundamental impact that I would expect from a publication in Nature Communications and I would recommend acceptance.

The strategy which the authors employ to discriminate between doubly bonded and diradical forms is by changing the electronic properties of the ancillary carbene donor – and in particular by using the cAAC family of ligands which are well known to offer strong pi acceptor properties, and which have been used in a number of previous systems to stabilize adjacent radical species. The resulting NHC and cAAC systems have been characterized very thoroughly – both computationally and experimentally - and the key magnetic properties of the diradical species have been examined in depth. As such, I have no significant comments to make concerning the scientific narrative of the paper, which reads very well.

The one minor omission – if one were being picky – would be some demonstration of the diradical character of 2 via onward reaction chemistry. This would be of particular interest given – for example – the role thought to be played by diradical forms in the chemistry of some of Power's dimetallyne systems. However, I would not regard this as being an essential addition for publication.

RESPONSE: We have thus far not been able to determine any clearcut reactivity of these molecules. This will be a subject of future work.

Two very minor points:

The description of the industrial use of olefins as being 'universal' is perhaps overstating the case! Perhaps 'widespread' or if greater impact is desired - 'critical' but 'universal' implies that ALL industries use olefins....

RESPONSE: We agree and have replaced the term.

The term 'captodatively' – is this the correct use of the term? Viehe's Acc Chem Res paper (ref 30) describes this as being the stabilization of radicals through the combined effect of donor and acceptor components on the radical centre (e.g. using alpha amino and cyano functions). The use of cAACs in stabilizing radical systems seems to be derived ostensibly from their pi acceptor characteristics and (as here) the consequent delocalization of spin density around the heterocycle.

RESPONSE: We realise that this term was confusing the way we used it in the text, as if you consider the unpaired electrons to be on the boron atoms then "captodative" does not apply. However, in the text we mention that the spin density is moved closer to the CAAC carbon atoms, which experience "captodative" stabilization from the pi-donor N and pi-acceptor B atoms. Thus the term is justified, and we have now clarified this in some places to remove the confusion.